

**Photoenhanced sulfates formation by the heterogeneous uptake of $SO_2$ on non-**
**photoactive mineral dust**
Chong Han*, Jiawei Ma, Wangjin Yang, Hongxing Yang
School of Metallurgy, Northeastern University, Shenyang, 110819, China
*Address correspondence to author: hanch@smm.neu.edu.cn
**Short summary.** We provide direct evidences that light prominently enhances the conversion
of $SO_2$ to sulfates on non-photoactive mineral dust, where $^3SO_2$ can act as a pivotal trigger to
generate sulfates. Photochemical sulfate formation depends on $H_2O$, $O_2$, and basicity of mineral
dust. It is suggested that the $SO_2$ photochemistry on non-photoactive mineral dust significantly
contributes to sulfates, highlighting previously unknown pathway to better explain the missing
sources of atmospheric sulfates.
**Abstract.** Heterogeneous uptake of $SO_2$ on mineral dust is a predominant formation pathway
of sulfates, whereas the contribution of photo-induced $SO_2$ oxidation to sulfates on the dust
interfaces still remains unclear. Here, we investigated heterogeneous photochemical reactions
of $SO_2$ on five mineral oxides ($SiO_2$, kaolinite, $Al_2O_3$, MgO, and CaO) without photocatalytic
activity. Light significantly enhanced the uptake of $SO_2$, and its enhancement effects negatively
depended on the basicity of mineral oxides. The initial uptake coefficient ($\gamma_{0,\,BET}$) and the
steady-state uptake coefficient ($\gamma_{s,\,BET}$) of $SO_2$ positively relied on light intensity, relative
humidity (RH) and $O_2$ content, while they exhibited a negative relationship with the initial $SO_2$
concentration. Rapid sulfate formation during photo-induced heterogeneous reactions of $SO_2$
with all mineral oxides was confirmed to be ubiquitous, and $H_2O$ and $O_2$ played the key roles
in the conversion of $SO_2$ to sulfates. Specially, $^3SO_2$ was suggested to be the trigger for
photochemical sulfate formation. Atmospheric implications supported a potential contribution
of interfacial $SO_2$ photochemistry on non-photoactive mineral dust to atmospheric sulfate
sources.




**Keywords:** $SO_2$; Sulfates; Non-photoactive mineral dust; Heterogeneous photochemistry



**1 Introduction**

As an important trace gas in the atmosphere, $SO_2$ is mainly emitted by volcanic eruption and fuel combustion. There is an uneven distribution of atmospheric $SO_2$ concentrations that show a distinctive seasonal and regional differentiation. Typical ratios of $SO_2$ in the troposphere are below 0.5 ppb for a clean weather in remote areas, rising to around several hundred ppb during the polluted days in urban regions(Ma et al., 2020). About half of $SO_2$ is oxidized to sulfates(He et al., 2012), which is one of the most significant compositions in fine particles. Sulfates can contribute greatly to the mass concentration of $PM_{2.5}$, with the mass of sulfates high up to 30%(Shao et al., 2019), especially in polluted regions where high-sulfur fuels are usually used(Olson et al., 2021). They significantly alter physicochemical properties of aerosols in terms of hygroscopicity, acidity and light absorption property(Chan and Chan, 2003; Cao et al., 2013; Lim et al., 2018). Sulfates also pose a human health risk through causing respiratory illness and cardiovascular(Shiraiwa et al., 2017). In addition, the deposition of sulfates leads to adverse effects on ecosystems via the acidification of soils and lakes(Golobokova et al., 2020). Therefore, the oxidation of $SO_2$ to form sulfates has attracted widespread attentions in the past decades.

The conversion of $SO_2$ to sulfates in the atmosphere usually occur by three ways: gas-phase oxidation of $SO_2$ by hydroxyl radicals (•OH) or Criegee intermediate radicals(Mauldin et al., 2012; Davis et al., 1979); aqueous-phase reaction of $SO_2$ with $O_3$, $H_2O_2$ or transition metal ions dissolved in cloud and fog droplets(Alexander et al., 2009; Herrmann et al., 2000; Liu et al., 2020a; Li et al., 2020); and heterogeneous $SO_2$ uptake on aerosols including authentic mineral dust, soot, inorganic ion and organic compounds(Adams et al., 2005; He et al., 2018a; Zhang et al., 2020a; Liu et al., 2020b). However, the oxidation of $SO_2$ in gas and aqueous phases fails to explain high sulfate concentrations under polluted conditions. Model simulation suggests

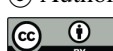



that the rapid sulfate formation can be attributed to the heterogeneous SO$_2$ uptake(Li et al.,
2017). A positive relationship between the fraction of sulfates and mineral dust in haze days
has been reported, implying that mineral dust may account for the formation of sulfates(Wang
et al., 2020a). Moreover, a large amount of sulfates was observed to be formed on the surface
of mineral dust after long-distance transport(Prospero, 1999). Thus, investigating the
heterogeneous oxidation of SO$_2$ on mineral dust is of fundamental importance to reveal large
missing sources of atmospheric sulfates in the haze periods.
Mineral dust, regarded as the dominant component of particulate matters in the atmosphere,
accounts for about 30%−60% mass fractions of global aerosols(Dentener et al., 1996; Peng et
al., 2012). It primarily contains SiO$_2$ (40%−80%), followed by Al$_2$O$_3$ (10%−15%), Fe$_2$O$_3$
(6%−13%), CaO (3%−10%), MgO (1%−7%) and TiO$_2$ (0.1%−5%)(Urupina et al., 2021;
Urupina et al., 2019; Usher et al., 2003). Mineral dust can provide active sites for adsorption
and reaction of gases. Up to now, the heterogeneous SO$_2$ uptake on authentic mineral aerosols
and model mineral oxides has been widely reported(Ma et al., 2019; Goodman et al., 2001;
Wang et al., 2018; Wang et al., 2020b), with various uptake coefficients ($\gamma$) of SO$_2$ varying
from $10^{-9}$ to $10^{-4}$(Urupina et al., 2019; Usher et al., 2002).
It was recognized that light could significantly enhance heterogeneous conversion of SO$_2$ to
sulfates on the surface of photocatalytic mineral dust(Chen et al., 2021; Li et al., 2019; Wang
et al., 2020b). Electron-hole pairs are produced via photo-induced electrons from the valence
band to the conduction band of photocatalytic metal oxides, and then react with H$_2$O and O$_2$ to
generate reactive oxygen species (ROS), such as •OH and •O$_2^-$(Chu et al., 2019). Sulfates are
produced by the heterogeneous reactions of SO$_2$ and ROS(Park and Jang, 2016; Park et al.,
2017; Langhammer et al., 2020; Bounechada et al., 2017). In particular, due to the large
abundance of non-photoactive mineral dust (more than 85% mass of total mineral dust in the
atmosphere) (Usher et al., 2003; Liu et al., 2022), revealing the photooxidation processes of
SO$_2$ on these mineral dust is of great importance to better reevaluate the sulfate formation on
aerosols in the global scale.
Hence, photochemical SO$_2$ uptake and sulfate formation on non-photoactive mineral oxides



were firstly investigated using a flow reactor and an *in situ* diffuse reflectance infrared Fourier
transform spectroscopy (DRIFTS). The $SO_2$ conversion to sulfates was examined under various
conditions, and the roles of light intensity, $SO_2$ concentration, $H_2O$, $O_2$ and basicity of mineral
oxides were determined. Reaction mechanisms and atmospheric implications were proposed,
highlighting a new and important pathway accounting for photochemical uptake of $SO_2$ to form
sulfates on the non-photoactive surfaces.

**2 Experimental methods**
**2.1 Materials**.
Analytical grade $SiO_2$ (Sinopharm Chemical Reagent Co., Ltd.), kaolinite (Sinopharm
Chemical Reagent Co., Ltd.), $Al_2O_3$ (Alfa Aesar), MgO (Sigma-Aldrich), and CaO (Sigma-
Aldrich) were used in the experiments. Through the nitrogen Brunauer-Emmett-Teller (BET)
physisorption analysis, their specific surface areas were detected to be 0.419, 6.407, 8.137,
10.948 and 6.944 $m^2 \ g^{-1}$, respectively. The ultraviolet-visible (UV-vis) light absorption spectra
of samples in the wavelength range of 300–800 nm were obtained by the Shimadzu UV-2550
spectrophotometer, as shown in Figure S1 of the Supporting Information. The solid powder
(0.2−5 g) was uniformly dispersed into 10.0 mL ethanol solution. The mixed liquid was poured
into a rectangle quartz sample dish (14.0 cm × 7.0 cm) and dried to form a solid coating in an
oven at 353 K for 10 h. $SO_2$ standard gas (50 ppm in $N_2$, Shenyang Air Liquide Co., LTD) and
high purity $N_2$ and $O_2$ (99.999 vol.%, Shenyang Air Liquide Co., LTD) were used as received.
The solid sample powder (0.2 g) was immersed into 10 ml deionized water (20 mg $ml^{-1}$), and
then the suspension was vigorously stirred for 10 min. The pH of $SiO_2$, kaolinite, $Al_2O_3$, MgO
and CaO suspension was measured to be 6.27, 6.58, 9.33, 10.61 and 12.72 using a pH meter,
respectively, which was employed to characterize the basicity of mineral oxides.
**2.2 Rectangular flow reactor**.
The uptake experiments of $SO_2$ on mineral dust were performed in a horizontal rectangular
flow reactor (26.0 cm length × 7.5 cm width × 2.0 cm height), which was depicted in Figure
S2. The reactor was made of quartz to allow the transmission of light. The temperature was



maintained at 298 K by circulating temperature-controlled water through the outer jacket of the
reactor. Synthetic air with a $N_2/O_2$ volume ratio of 4:1 was introduced into the flow reactor,
and its total flow rate was 1000 mL min$^{-1}$. The Reynolds number ($Re$) was calculated to be
14.42 ($Re < 200$), indicating a laminar flow state. $SO_2$ with high purity $N_2$ (100 mL min$^{-1}$) as
carrier gas were introduced into the reactor through a movable T-shaped injector equipped with
six exit holes of 0.2 mm diameter, so that the gas could be uniformly distributed over the width
of the reactor. The $SO_2$ concentration was 40−200 ppb and measured with a $SO_2$ analyzer
(Thermo 43i). Wet $N_2$ generated with a bubbler containing deionized water was introduced by
two parallel inlets on the side of T-shaped injector. Relative humidity (RH, 10%−75%) was
controlled by regulating the ratio of dry $N_2$ to wet $N_2$ and measured via a hygrometer (Center
314). The equivalent layer numbers of water on surface was 0.9−4.0 according to the Brunauer-
Emmett-Teller (BET) model(Sumner et al., 2004), and the thickness of the film of adsorbed
water varied between 2.7−12 nm at RH=10%−75%. There were three equally spaced exhaust
ports to mitigate the outlet turbulence. A xenon lamp (CEL-LAX500, China Education Au-light
Co., Ltd) was used to simulate sunlight and vertically located above the reactor. A filter was
placed on the reactor to remove the light with wavelengths shorter than 300 nm. The spectrum
irradiance of the xenon lamp was displayed in Figure S3 and measured using a calibrated
spectroradiometer (ULS2048CL-EVO, Avantes). The spectral irradiance was measured inside
the reactor, after passing the water cooling and in the absence of a sample. The total irradiance
($0−7.93 \times 10^{16}$ photons cm$^{-2}$ s$^{-1}$) on the coating can be adjusted by varying the distance of the
xenon lamp to the reactor.
**2.3 Uptake coefficient of $SO_2$**.
The heterogeneous reaction kinetics of $SO_2$ with mineral dust can be described by a pseudo-
first-order reaction. $SiO_2$ was taken as an example, and Figure S4 showed a linear relationship
between the natural logarithm of the $SO_2$ concentration and the reaction time. The apparent rate
constant ($k_{obs, SiO_2}$) of $SO_2$ with $SiO_2$ can be calculated using the equation 1,
$$\frac{\ln(C_0/C_t)}{t} = k_{obs, SiO_2} \quad (1)$$
where $C_0$ and $C_t$ (ppb) are the initial $SO_2$ concentration and the $SO_2$ concentration at the



reaction time $t$, respectively. The loss of $SO_2$ on the internal wall of the reactor occurred in
blank experiments (Figure S5), which should be deducted for the $\gamma$ calculation. Assuming that
the wall loss was constant in the experiments with and without samples, the geometric uptake
coefficient ($\gamma_{geo}$) was determined by the equation 2(Knopf et al., 2007),
$$\gamma_{geo} = \frac{4V(k_{obs,\,SiO_2} - k_{obs,\,wall})}{S\omega} \quad (2)$$
where $k_{obs,\,SiO_2}$ and $k_{obs,\,wall}$ are the apparent rate constants measured with and without $SiO_2$
samples, respectively; $V$ (m$^3$), $S$ (m$^2$) and $\omega$ (m s$^{-1}$) are the volume of the rectangular reactor,
the surface area of the sample dish, and the mean molecular speed of $SO_2$, respectively.

The uptake process of $SO_2$ on $SiO_2$ depended on the reaction of $SO_2$ with $SiO_2$ and the mass

transport of $SO_2$ to the surface. It can be expressed with the equation 3,
$$k'_{r,\,SiO_2} = \left[\frac{1}{k_{r,\,SiO_2}} - \frac{a}{N_uD}\right]^{-1} \quad (3)$$
where $k_{r,\,SiO_2} = k_{obs,\,SiO_2} - k_{obs,\,wall}$; $k'_{r,\,SiO_2}$ is the reaction rate constant of $SO_2$ accounting for the
diffusion effect; $D$ (cm$^2$ s$^{-1}$) is the diffusion coefficient of $SO_2$ in air; $a$ (cm) is one half height
of the flow reactor; $N_u$ is the Nusselt numbers obtained with a calculation method from Solbrig
and Gidaspow(Solbrig and Gidaspow, 1967), which represents the mass transport. Then, the
corrected $\gamma$ can be calculated by inserting the equation 3 into the equation 2. In our experiments,
the correction for $\gamma$ was estimated to be approximate 10%. Initial uptake coefficients ($\gamma_0$) and
steady-state uptake coefficients ($\gamma_s$) were calculated by averaging the signals within the 1.0 and
40−60 min reaction time, respectively.

To understand the diffusion depth of $SO_2$ and determine the interaction of $SO_2$ with the

underlying layers of $SiO_2$, the uptake of $SO_2$ as a function of the $SiO_2$ mass under irradiation
was shown in Figure S6. The $\gamma$ exhibited a linear increase in the $SiO_2$ mass range of 0.05−2.0
g, while it remained unchanged at the $SiO_2$ mass > 3.0 g. Therefore, the uptake coefficient of
$SO_2$ in the linear regions was normalized using the BET surface area of $SiO_2$ by the equation
4(Brunauer et al., 1938),
$$\gamma_{BET} = \frac{S_{geo} \times \gamma_{geo}}{S_{BET} \times m_{SiO_2}} \quad (4)$$




166 where $\gamma_{BET}$ is the SO$_2$ uptake coefficient normalized to the BET surface area; $S_{geo}$ (m$^2$) is the

167 geometric area of the sample dish; $S_{BET}$ (m$^2$) is the BET surface area of SiO$_2$; $m_{SiO_2}$ (g) is the

168 SiO$_2$ mass. The same method was also used to calculate the uptake coefficients of SO$_2$ on other

169 mineral oxides.

170 **2.4 In Situ DRIFTS analysis**.

171 The changes in the chemical compositions on mineral oxides in the SO$_2$ uptake process were

172 investigated by *in situ* diffuse reflectance Fourier-transform infrared spectroscopy (DRIFTS),

173 which was recorded using the Fourier transform infrared (FTIR) spectrometer (Thermo Nicolet

174 iS50) equipped with a mercury cadmium telluride (MCT) detector. About 14 mg mineral oxides

175 was placed into a stainless-steel cup inside the reaction cell. To remove adsorbed impurities,

176 SiO$_2$ was purged with a 150 mL min$^{-1}$ airflow (N$_2$/O$_2$ volume ratio = 4:1) at RH=40% for 1 h.

177 Then, a background spectrum of unreacted samples was collected. SO$_2$ (2 ppm) was introduced

178 into the reaction cell, and the IR spectra was recorded as a function of time at a resolution of 4

179 cm$^{-1}$ by averaging 100 scans. The light from the xenon lamp (500 W) with a total irradiance of

180 $3.22 \times 10^{16}$ photons cm$^{-2}$ s$^{-1}$ was transmitted into the DRIFTS reaction cell via a fiber.

182 **3 Results and discussion**

183 **3.1 Photo-enhanced uptake of SO$_2$**.

184 Acting as the most abundant mineral oxides, SiO$_2$ was firstly used to investigate the uptake

185 behaviors of SO$_2$. As shown in Figure 1A, when SO$_2$ (80 ppb) was exposed to SiO$_2$ in the dark,

186 the SO$_2$ concentration decreased to 70 ppb, and then it quickly increased and reached the steady

187 state after 20 min. Upon exposure to SiO$_2$ under irradiation, the SO$_2$ concentration exhibited a

188 greater drop than that in the dark. The deactivation of SO$_2$ uptake on SiO$_2$ was very slight after

189 20 mins under irradiation. These suggest that light can significantly promote the heterogeneous

190 reaction of SO$_2$ on SiO$_2$. When SO$_2$ didn't contact with SiO$_2$, its concentration recovered

191 rapidly. The desorption of SO$_2$ was observed when SO$_2$ was isolated from SiO$_2$ in the dark and

192 under irradiation, indicating that the physical adsorption partially contributed to the SO$_2$ loss

193 during the photochemical process.



The uptake coefficients of $SO_2$ on $SiO_2$ as a function of irradiation intensity were shown in
Figure 1B. The errors in all figures are the standard deviations of three repetitive experiments.
Both $\gamma_{0,\,BET}$ and $\gamma_{s,\,BET}$ displayed a well linear relationship with the irradiation intensity, further
confirming the photochemical nature for the reactions of $SO_2$ on $SiO_2$. In particular, $\gamma_{0,\,BET}$ and
$\gamma_{s,\,BET}$ on $SiO_2$ under simulated solar irradiation was comparable with those ($10^{-7}-10^{-6}$) on
Gobi Desert dust (GDD) and Arizona Test Dust (ATD) under UV irradiation, which contained
photocatalytic metal oxides(Park et al., 2017). As for the $SO_2$ uptake on $TiO_2$, $\gamma_{0,\,BET}$ and $\gamma_{s,\,BET}$
were measured to be $10^{-6}$ and $10^{-7}$, respectively, by using the flow tube(Ma et al., 2019), which
were similar to our results. Usher et al.(2022) reported a larger $\gamma_{BET}$ ($10^{-4}$) on $TiO_2$ using a
Knudsen cell reactor. This difference should be ascribed to the variation in the pressure of
Knudsen cell (high vacuum) and flow tube reactor (ambient pressure).
Figure 1C shows the evolution of $\gamma_{0,\,BET}$ and $\gamma_{s,\,BET}$ at different $SO_2$ concentrations under
irradiation. An inverse dependence of $\gamma_{0,\,BET}$ and $\gamma_{s,\,BET}$ on the $SO_2$ concentration was observed,
meaning that both initial and steady-state uptake reactions were lower efficient at higher $SO_2$
concentrations. The uptake of gases on the solid surfaces usually follows the Langmuir-
Hinshelwood (L-H) mechanism(Ammann et al., 2003; Zhang et al., 2020b), suggesting that
gaseous molecules are quickly absorbed on the surfaces, and then the reactions occur among
the absorbed species. Assuming that the adsorption of $SO_2$ on $SiO_2$ is in accord with the
Langmuir isotherm, the dependence of $\gamma$ on the $SO_2$ concentration can be described by the
equation 5(Zhang et al., 2020b),
$$\gamma = \frac{(4V/S\omega)k[SiO_2]_T K_{SO_2}}{1+K_{SO_2}[SO_2]_g} \qquad (5)$$
where $[SO_2]_g$ is the concentration of gaseous $SO_2$; $[SiO_2]_T$ is the total number of active sites
on $SiO_2$; $k$ is the reaction rate constant of $SO_2$ absorbed on $SiO_2$; $K_{SO_2}$ represents the Langmuir
adsorption constant of $SO_2$. Because the $SiO_2$ mass remained constant during the reaction, the
equation 5 can be written as the equation 6,



$\gamma = \dfrac{a}{1 + K_{SO_2}[SO_2]_g}$          (6)
where $a = (4V/S\omega)k[SiO_2]K_{SO_2}$. As shown in Figure 1C, the equation 6 can well describe the
correlation of the $SO_2$ uptake coefficient with the $SO_2$ concentration, suggesting that the L-H
mechanism can explain the influence of the $SO_2$ concentration on $\gamma_{0,\,BET}$ and $\gamma_{s,\,BET}$.

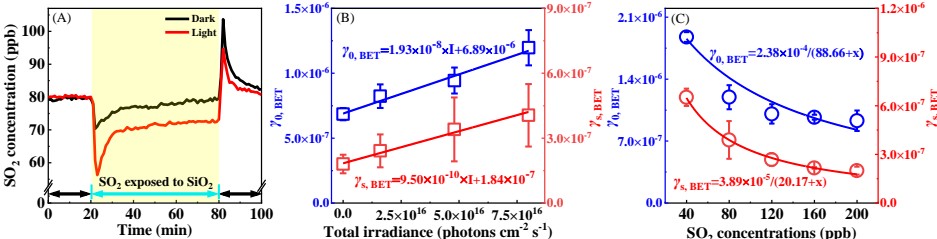

**Figure 1.** (A) The temporal variation of the $SO_2$ concentration on $SiO_2$ in the dark and under
irradiation (250 W m$^{-2}$); The background changes of the $SO_2$ concentration in the blank
reactor have been deducted. (B) The $\gamma_{0,\,BET}$ and $\gamma_{s,\,BET}$ of $SO_2$ on $SiO_2$ as a function of the
light intensity. (C) The $\gamma_{0,\,BET}$ and $\gamma_{s,\,BET}$ of $SO_2$ on $SiO_2$ at different $SO_2$ concentrations under
irradiation (250 W m$^{-2}$); The fitting lines for $\gamma_{0,\,BET}$ and $\gamma_{s,\,BET}$ were based on the Langmuir-
Hinshelwood mechanism using the equation 6. Reaction conditions: $SiO_2$ mass of 0.2 g,

temperature of 298 K, RH of 40% and $O_2$ content of 20%.


**3.2 Photo-induced formation of sulfates by the SO₂ uptake.**
To investigate the products formed on $SiO_2$, *in situ* DRIFTS spectra were recorded, as shown
in Figure 2. The band at 1359 cm$^{-1}$ was assigned to physically-adsorbed $SO_2$ on $SiO_2$(Urupina
et al., 2019). The bidentate sulfate and bisulfate contributed to the bands at 1260 and 1229
cm$^{-1}$(Urupina et al., 2019; Yang et al., 2020), respectively. The bands at 1074 and 1038 cm$^{-1}$
may be related to the monodentate sulfite(Yang et al., 2019; Wang et al., 2019). It was noted
that the intensity of physically-absorbed $SO_2$ (1359 cm$^{-1}$) under irradiation was lower than that
in the dark (Figure S7), which may be ascribed to further conversion of $SO_2$ absorbed on $SiO_2$
under irradiation. Especially, the sulfate bands (1260 and 1229 cm$^{-1}$) appeared under irradiation,



while they were not observed in the dark. This clearly confirms the crucial role of light in the
heterogeneous conversion of $SO_2$ to sulfates on $SiO_2$. The bands of sulfites (1038 and 974 cm$^{-1}$)
under dark and irradiation conditions became more apparent with the reaction time, suggesting
the continuous formation of sulfites.

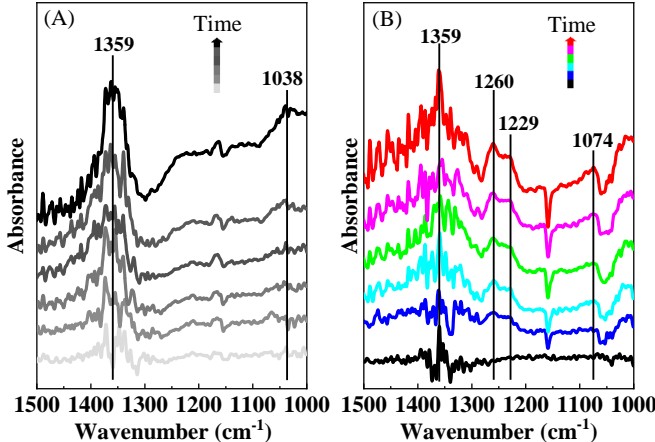


**Figure 2.** *In situ* DRIFTS spectra of $SiO_2$ during the uptake process of $SO_2$ (2 ppm) in the
dark (A) and under irradiation (B). Reaction conditions: RH of 40%, temperature of 298 K
and $O_2$ content of 20%.

**3.3 Key roles of $H_2O$ and $O_2$ in photochemical conversion of $SO_2$ to sulfates**.
Figure S8A shows temporal variations of the $SO_2$ concentration in the reaction with $SiO_2$ at
RH=10% and 60% under irradiation. The uptake of $SO_2$ was very weak at RH=10%, whereas
it was obvious at RH=60%. Moreover, $H_2O$ markedly prolonged the time to reach the steady-
state uptake of $SO_2$. This definitely determines that $H_2O$ plays a distinct enhancement role in
the photochemical uptake of $SO_2$. As shown in Figure 3A, $\gamma_{0,\,BET}$ had a continuous increase
from $(1.20 \pm 0.04) \times 10^{-7}$ to $(1.54 \pm 0.07) \times 10^{-6}$ with increasing the RH in the 10%−60% range,
but it decreased to $(1.05 \pm 0.09) \times 10^{-6}$ at RH=75%. The $\gamma_{s,\,BET}$ linearly depended on the RH,
and linear fitting to $\gamma_{s,\,BET}$ versus RH yielded the equation $\gamma_{s,\,BET}=1.31\times10^{-8}\times RH-1.02\times10^{-7}$.
Adsorbed $H_2O$ promoted the hydration and dissociation of $SO_2$(Huang et al., 2015), and it may





generate reactive oxygen species (ROS) such as •OH or $HO_2$ radicals to oxidize $SO_2$ under
irradiation(Li et al., 2020; Ma et al., 2019), which would lead to positive effects of RH on the
$SO_2$ uptake. Adsorbed $H_2O$ also occupied adsorptive and active sites on the surface, and
produced the competition with $SO_2$. When this competitive role was dominated, the uptake of
$SO_2$ would be hindered.

The DRIFTS spectra of $SiO_2$ during the $SO_2$ uptake at different RHs are shown in Figure

S9A. The band intensities of sulfates (1260 and 1229 $cm^{-1}$) at RH=60% were greatly stronger
than those at RH=10%, suggesting that $H_2O$ significantly promotes the sulfate formation. To
further investigate the influence of $H_2O$ on the sulfate formation, the integrated area of sulfates
in the DRIFTS spectra (1289−1202 $cm^{-1}$) as a function of the time at different RHs is illustrated
in Figure 3B. Sulfates exhibited a fast formation in the initial 30 min at any RH, and then they
were continuously generated at a relatively slow rate. Absorptive sites for $SO_2$ can be blocked
because of the accumulation of $H_2O$ and products (sulfites and sulfates), resulting in the gradual
deactivation of the surface. It was noted that sulfates had a more distinct formation trend with
increasing the RH, revealing that $H_2O$ can act as an important participator in the production of
sulfates by the photochemical uptake of $SO_2$ on $SiO_2$.

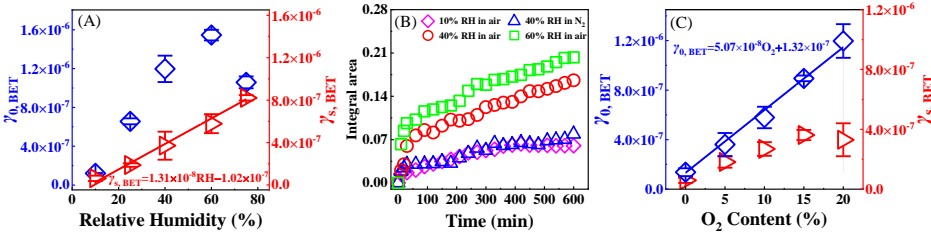


**Figure 3.** (A) The dependence of $\gamma_{0,\,BET}$ and $\gamma_{s,\,BET}$ on RH. (B) Integrated area of sulfates in
*in situ* DRIFTS spectra (1289−1202 $cm^{-1}$) as a function of time. (C) The dependence of
$\gamma_{0,\,BET}$ and $\gamma_{s,\,BET}$ on $O_2$. Reaction conditions: $SiO_2$ mass of 0.2 g, irradiation intensity of 250

W $m^{-2}$, temperature of 298 K, $O_2$ content of 20% for (A) and RH of 40% for (B).


Figure S8B displays effects of $O_2$ on the photochemical uptake of $SO_2$ on $SiO_2$. Negligible

$SO_2$ uptake occurred in $N_2$, while there was a significant decrease in the $SO_2$ concentration in





284 air. The $\gamma_{0,\,BET}$ greatly increased from $(1.37 \pm 0.45) \times 10^{-7}$ in $N_2$ to $(1.19 \pm 0.13) \times 10^{-6}$ in 20%

285 $O_2$ (Figure 3C), confirming that $O_2$ was involved in the photochemical reaction of $SO_2$ on $SiO_2$.

286 The $\gamma_{s,\,BET}$ displayed different dependence behaviors on $O_2$. It exhibited an increase from (7.10

287 $\pm 2.85) \times 10^{-8}$ in $N_2$ to $(4.37 \pm 0.58) \times 10^{-7}$ in 15% $O_2$, whereas it remained unchanged in 20%

288 $O_2$.

289  DRIFTS spectra of $SiO_2$ during the $SO_2$ uptake in $N_2$ and air was compared in Figure S9B.

290 In both air and $N_2$, the bands of absorbed $SO_2$ (1359 cm$^{-1}$), sulfates (1260 and 1229 cm$^{-1}$) and

291 sulfites (1074 cm$^{-1}$) appeared. Nevertheless, their intensities in $N_2$ were weaker than those in

292 air. According to the integrated area of sulfates in the DRIFTS spectra (1289−1202 cm$^{-1}$) as a

293 function of time, the formation trends of sulfates were similar in $N_2$ and air (Figure 3B), while

294 the sulfate formation rate in $N_2$ was obviously lower than that in air, meaning that $O_2$ enhanced

295 the sulfate production. It was reported that the production rate of sulfates from the $SO_2$ uptake

296 on $TiO_2$ and by the photolysis of nitrates under UV irradiation in $N_2$ was also smaller than that

297 in air(Ma et al., 2019; Gen et al., 2019b). In addition, it was noted that sulfates can be generated

298 in $N_2$, meaning that $O_2$ was not necessary and some pathways contributed to sulfates without

299 $O_2$.

300

301 **3.4 Ubiquitously photoenhanced conversion of SO₂ to sulfates**.

302  To better assess the potential for photochemical conversion of $SO_2$ to sulfates, the $SO_2$ uptake

303 experiments were further performed for typical mineral oxides without photocatalytic activity.

304 As displayed in Figure S10, more obvious uptake behaviors of $SO_2$ on kaolinite, $Al_2O_3$, MgO

305 and CaO were observed under irradiation when compared to those in the dark. Figure 4A shows

306 that there was the largest $\gamma_{s,\,BET}$ for CaO among five minerals, and $\gamma_{s,\,BET}$ positively depended

307 on the basicity (pH) of mineral oxides. Basic oxides generally contains more surface hydroxyls,

308 which is in favor of sulfite and sulfate formation to enhance the heterogeneous uptake of

309 $SO_2$(Zhang et al., 2006). Moreover, the ratios of steady-state uptake coefficients under

310 irradiation to those in the dark ($\gamma_{s,\,Light}/\gamma_{s,\,Dark}$) were larger than 1.0 for all mineral oxides





(Figure 4B), determining a general enhancement role of light in the SO$_2$ uptake. However, the
$\gamma_{Light}/\gamma_{Dark}$ had smaller values with an increase in the basicity, suggesting that the promotion
effect of the light was less remarkable for basic oxides.

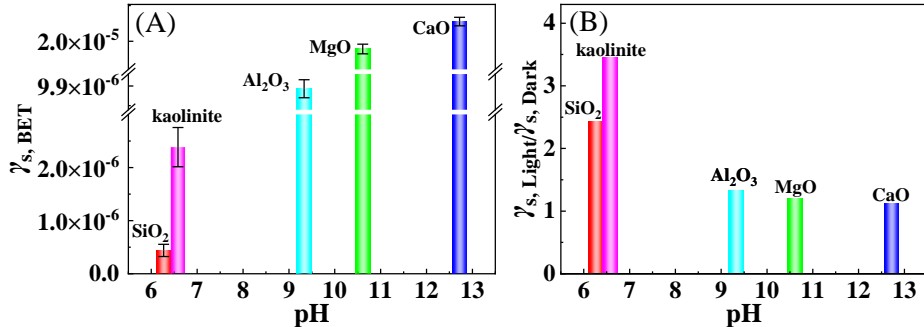

**Figure 4.** (A) The dependence of $\gamma_{s, BET}$ under irradiation on the basicity (pH) of mineral
oxides. (B) The ratios of steady-state uptake coefficients under irradiation to those in the dark
($\gamma_{s, Light}/\gamma_{s, Dark}$) for different mineral oxides. Reaction conditions: mineral oxides mass of 0.2
g, irradiation intensity of 250 W m$^{-2}$, temperature of 298 K, RH of 40% and O$_2$ content of

319 20%.


As shown in Figure 5A and B, no formation of sulfates was found for kaolinite in the dark,
while bisulfates (1220 cm$^{-1}$) were observed under irradiation. Compared to weaker peaks of
sulfates (1200 and 1260 cm$^{-1}$) for Al$_2$O$_3$ in the dark (Figure 5C), a stronger band of bisulfates
appeared at 1220 cm$^{-1}$ under irradiation (Figure 5D). By contrast to the generation of sulfates
for kaolinite and Al$_2$O$_3$, both sulfites and sulfates formations were observed for MgO and CaO
(Figure 5E-H). Sulfites were dominant in the dark, as shown by the peaks at 966 and 1020 cm$^{-1}$
for MgO and 943 cm$^{-1}$ for CaO, whereas the sulfate formation was significantly enhanced
under irradiation according to peak intensities at 1163 cm$^{-1}$ for MgO and 1137 cm$^{-1}$ for CaO.
It should be noted that these mineral oxides were non-photoactive because of their poor light
absorption property (Figure S1). Nevertheless, it was very surprised that the light can greatly
promote the formation of sulfates via the SO$_2$ uptake process on mineral oxides without





photocatalytic activity, which was strongly suggested to be a new and important finding for
atmospheric sulfate sources.

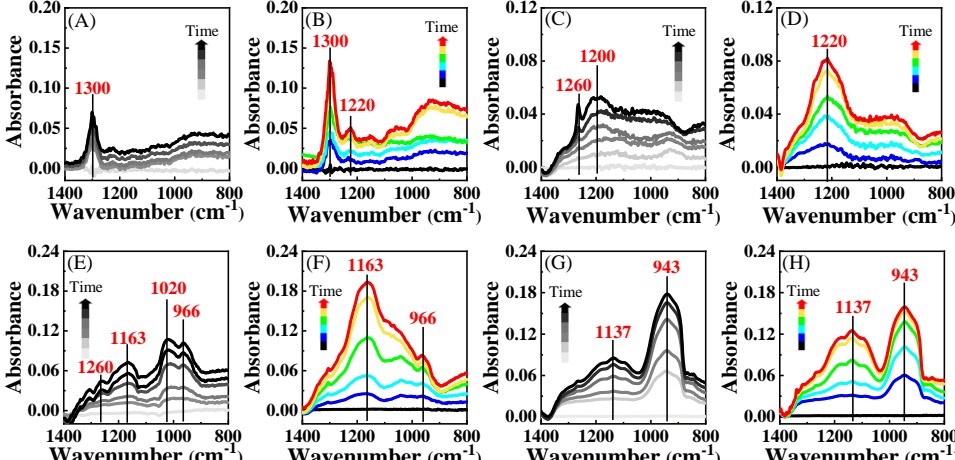

**Figure 5.** *In situ* DRIFTS spectra of kaolinite (A and B), $Al_2O_3$ (C and D), MgO (E and F),
CaO (G and H) during the uptake process of $SO_2$ (2 ppm) for 600 min in the dark (black
lines) and under irradiation (colorful lines). Reaction conditions: RH of 40%, temperature of
298 K and $O_2$ content of 20%.

**3.5 Conversion mechanisms of $SO_2$ to sulfates**.
Heterogeneous photochemical reaction mechanisms of $SO_2$ on non-photoactive mineral dust
were proposed in light of experimental observations (Figure 6). Gaseous $SO_2$ was adsorbed on
the surface (R1), and then reacted with $H_2O$ to form sulfites (R2). Under irradiation, adsorbed
$SO_2$ accepted photons to form its singlet states ($^1SO_2$) and triplet states ($^3SO_2$) (R3−5)
(Sidebottom et al., 1972; Martins-Costa et al., 2018). The reaction between $^3SO_2$ and $H_2O$
resulted in the formation of HOSO• and •OH (R6), which can combine with $SO_2$ to produce
HOSO$_2$• (R7). HOSO• and HOSO$_2$• can be transformed into $SO_3$, which reacted with $H_2O$ to
drove the sulfate formation (R8 and R9). The interaction between $^3SO_2$ and $O_2$ may also
generate $SO_3$ directly, which would be converted to sulfates subsequently (R10). Theoretical
calculations suggested that the multistep reactions between $^3SO_2$ with $H_2O$ and $O_2$ had small
energy barriers or were barrier-free(Gong et al., 2022), which could enhance the generation of



ROS and the transformation of S(IV) to S(VI). As displayed by R11 and R12, $SO_2$ and $H_2SO_3$
adsorbed on the surface may be oxidized to form sulfates via the reactions with ROS including
•O, •OH or $HO_2$• , which were produced in R6 and R8-10. In addition, gaseous $SO_2$ could be
dissolved into adsorbed $H_2O$ to generate bisulfites, which would be finally converted to sulfates
by ROS (R13) (Urupina et al., 2019). As displayed in Figure S11A, the IR peaks of sulfates
were not observed when tris (2,2′-bipyridine) ruthenium dihydrochloride ($Ru(bpy)_3(Cl)_2$) was
employed as the quencher of $^3SO_2$(Bulgakov and Safonova, 1996). The peaks were assigned to
the vibrations of excited $Ru(bpy)_3(Cl)_2$(Mukuta et al., 2014). This definitely proves that $^3SO_2$
is the key trigger for the sulfate formation. $NaHCO_3$ can be used as an efficient•OH scavenger
to determine the role of •OH deriving from the $^3SO_2$ reactions(Gen et al., 2019a). Figure S11B
shows that the peaks of sulfates were obviously weaker in the presence of $NaHCO_3$, confirming
the dominant contribution of •OH formed in R6 and R9 to the formation of sulfates.
Several photochemical mechanisms have been reported to explain the sulfate formation via
the $SO_2$ uptake on various surfaces. Photoactive mineral oxides (such as $TiO_2$, $F_2O_3$ and ZnO)
can accept photons to produce electron-hole pairs, which generated ROS for the conversion of
$SO_2$ to sulfates(Ma et al., 2019; Li et al., 2019; Wang et al., 2020b). For example, •OH and
$HO_2$•, generated from the reaction of hole with $H_2O$ and electron with $O_2$, respectively, can act
as oxidizing agents for the reaction with $SO_2$(Ma et al., 2019). Similarly, the reaction of $SO_2$
with photo-induced •OH obviously enhanced the formation of sulfate on diesel soot and actual
$PM_{2.5}$(Zhang et al., 2022; Zhang et al., 2020c). $NO_2$ and $NO_2^-$/$HNO_2$ can be formed in the
nitrates photolysis, and primarily contributed to the oxidation of $SO_2$ to sulfates on nitrates(Gen
et al., 2019b; Gen et al., 2019a). Theoretically, the mechanism proposed in this study should
also occur on photo-excited substrates. Taking $TiO_2$ as an example, $SO_2$ competed with $TiO_2$
for photons, and the production efficiency of $^3SO_2$ and excited state of $TiO_2$ ($TiO_2$*) depended
on their light absorption properties. Meanwhile, $^3SO_2$ had a competition electron-hole pairs
generated from $TiO_2$* for $O_2$ and $H_2O$. Thus, the dominant mechanism for the $SO_2$ uptake on
$TiO_2$ should be related to light absorption properties of precursors and the reactivity for $^3SO_2$
and $TiO_2$* to $O_2$ and $H_2O$. By contrast, all mineral oxides used here cannot be excited under



irradiation according to their light absorption spectra (Figure S1). Nevertheless, $SO_2$ adsorbed
on mineral oxides can absorb the ultraviolet radiation (290−400 nm) to form the excited states
of $SO_2$ ($SO_2^*$)(Kroll et al., 2018), which subsequently reacted with $H_2O$ and $O_2$, finally
converting $SO_2$ to sulfates. It means that any surfaces, providing absorptive sites for $SO_2$, can
significantly enhance the photooxidation of $SO_2$ to sulfates.

**Figure 6.** The proposed photochemical conversion mechanisms of $SO_2$ to sulfates on non-

photoactive mineral dust.


**4 Atmospheric implications**
The lifetime ($\tau$) for photochemical loss of $SO_2$ on mineral dust was given using the equation

7,

$$\tau = \frac{4}{\gamma \omega A} \quad (7)$$
where $\gamma$ and $\omega$ are the uptake coefficient and the mean molecular speed of $SO_2$, respectively; $A$
is the surface area density of mineral dust, and it is estimated to be (1.4–4.8) × $10^{-5}$ cm$^2$
cm$^{-3}$(Zhang et al.,2019; He et al., 2018b). In this work, $\gamma_{s, BET}$ of $SO_2$ on several mineral oxides
were measured to be from 4.39 × $10^{-7}$ to 3.45 × $10^{-5}$ under conditions with $SO_2$ concentration
of 40 ppb, irradiation intensity of 250 W m$^{-2}$ and RH of 40%. Thus, the $\tau$ of $SO_2$ with respect



to the photooxidation on mineral dust was calculated to be 0.9–240 days, which was shorter
than that (54 years) for the photochemical uptake of $SO_2$ on $TiO_2$ and the corresponding one
(346 days) for the heterogeneous oxidation of $SO_2$ on ATD in the presence of nitrates(Ma et al.,
2019; Zhang et al., 2019). It should be pointed out that the content of $TiO_2$ in mineral dust was
only about 1%, and thus the surface area density of $TiO_2$ was about $10^{-7}$ $cm^2$ $cm^{-3}$, leading to
a longer $\tau$ (54 years) for the loss of $SO_2$ on $TiO_2$(Ma et al., 2019). It was comparable to the
lifetime (3.6−20 days) of $SO_2$ for the gas-phase reaction with •OH at a concentration of ~$10^{-6}$
molecules $cm^{-3}$(Huang et al., 2015; Zhang et al., 2019). Therefore, the photochemical process
with the excited state $SO_2$ acting as a driver on mineral dust was an important pathway for the
$SO_2$ sink in the atmosphere.

Sulfates show significant influences on the atmosphere, such as an important contributor to

the haze formation, affecting the activity of aerosols acting as cloud condensation nuclei (CCN)
and ice nuclei (IN), and modifying optical property and acidity of aerosols. A sulfate formation
rate ($R$) can be obtained using $\gamma$ by the equation 8(Cheng et al., 2016),
$$R = \frac{d\left[SO_4^{2-}\right]}{dt} = \left[\frac{R_p}{D} + \frac{4}{\gamma\omega}\right]^{-1} A\left[SO_2\right] \qquad (8)$$
where $R_P$ is the radius of mineral dust, which can be estimated using the equation 9(Li et al.,

2020),

$R_P$ = (0.254 × [$PM_{2.5}$]/(μg $m^{-3}$) + 10.259) × $10^{-9}$ m    (9)
where [$PM_{2.5}$] was average $PM_{2.5}$ mass concentration, and 300 μg $m^{-3}$ was used for the polluted
periods in typical China cities(Li et al., 2020; Guo et al., 2014). It was assumed that mineral
dust accounted for 50% mass of $PM_{2.5}$(Tohidi et al., 2022), and the mass fraction of $SiO_2$, $Al_2O_3$,
MgO, and CaO in mineral dust was 60%, 12.5%, 4% and 6.5%, respectively(Urupina et al.,
2021; Urupina et al., 2019; Usher et al., 2003). Thus, $R$ was determined to be 2.15 μg $m^{-3}$ $h^{-1}$
according to $\gamma_{s, BET}$ under environmental conditions above. Table S1 summarizes sulfate
formation rates from various $SO_2$ oxidation pathways, including gas-phase reaction with
•OH(Xue et al., 2016), aqueous-phase oxidation by dissolved $NO_2$, $H_2O_2$ and TMI
catalysis(Cheng et al., 2016; Liu et al., 2020a; Ye et al., 2021), and heterogeneous



photochemistry on the surfaces of nitrates(Gen et al., 2019a), brown carbon(Liu et al., 2020b),
black carbon and $PM_{2.5}$(Zhang et al., 2022; Zhang et al., 2020c). It was clearly noted that sulfate
formation rates on non-photocatalytic mineral oxides under simulated sunlight were
comparable with those (0.001–10 μg m$^{-3}$ h$^{-1}$) for various pathways above, which may explain
the missing sulfate sources in the atmosphere. Accordingly, this new sulfate pathway should be
well taken into the full consideration in further field observation and model simulation studies
to better quantify atmospheric sulfate formation.

**Author contributions**
CH and JM designed and conducted experiments; CH and JM analyzed the data and prepared
the paper with contributions from WY and HY. CH supervised the project.

**Competing interests**
The authors declare that they have no conflict of interest.

**Acknowledgements**
This work was supported by the National Natural Science Foundation of China [grant number
42077198], the LiaoNing Revitalization Talents Program [grant number XLYC1907185], and
the Fundamental Research Funds for the Central Universities [grant numbers N2025011].

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
