# Peer review of "Photoenhanced sulfates formation by the heterogeneous uptake of $SO_2$ on non- photoactive mineral dust"

_EGUsphere, 2023_

## Author Comment (AC1)

Dear professor Thorsten Bartels-Rausch,

Thank you for your decision to allow us to revise our manuscript for publication in *Atmospheric Chemistry and Physics*. Below is a point by point response to the comments.

**Reviewer 1:**

The heterogeneous conversion of $SO_2$ to sulfates on non-photoactive surfaces was well investigated in this paper. The authors reported that light can enhance the $SO_2$ uptake and sulfate formation on non-photoactive surfaces ($SiO_2$, $Al_2O_3$, kaolinite, CaO and MgO). The sulfate formation pathway involving in the participation of $O_2$ and $H_2O$ was proposed. This is a novel topic since the previous studies generally focused on the sulfate formation on photoactive surfaces, such as $TiO_2$, $Fe_2O_3$ and etc.. This study highlighted a new pathway that contributed to the source of atmospheric sulfates. The paper was organized with logic, and the conclusion is convincing. I would recommend this manuscript for publication in ACP after a minor revision.

**Re:** Thank you for your comments.

Specific Comments

Lines 38-39: The two sentences "sulfates, which is one of the most significant compositions in fine particles." and "Sulfates can contribute greatly to the mass concentration of $PM_{2.5}$" have the same meaning. I suggest simplifying the second sentence into "the mass of sulfates in $PM_{2.5}$ is high up to 30%".

**Re:** Thank you. According to the suggestion from reviewers, the sentence has been modified into "The mass fraction of sulfates in $PM_{2.5}$ is high up to 30%" in Lines 38-39 in the revised manuscript.

Line 77: "and" should be modified into "with".

**Re:** Thank you. This has been modified in Line 77 in the revised manuscript.

Line 97: The details for measuring the light absorption spectra of samples should be given.

**Re:** Thank you. With $BaSO_4$ used as the reference, the ultraviolet-visible (UV-vis) light absorption spectra of samples (Figure S1) in the wavelength range of 300–800 nm were obtained by the Shimadzu UV-2550 spectrophotometer, which was equipped with diffuse reflection integrating sphere attachment. This description has been added in Lines 97-100 in the revised manuscript.

Line 104: "ml" should be modified into "mL".

**Re:** Thank you. This has been modified in Line 105 in the revised manuscript.

Lines 109-110: Rectangular flow reactor was not a conventional reactor. Was this reactor used in the previous paper? Or the feasibility of this reactor was verified before?

**Re:** Thank you. In a previous study, a similar rectangular flow reactor was designed and the feasibility of the reactor has been verified (Knopf et al., *J. Phys. Chem. A*, 2007, 111, 11021-11032). This description has been added in Lines 112-113 in the revised manuscript.

Lines 131, 225, 228, 279, 318 and 397: The unit of light intensity (photons $cm^{-2}$ $s^{-1}$ or $W$ $m^{-2}$) should be unified.

**Re:** Thank you. The unit of light intensity has been unified to "photons $cm^{-2}$ $s^{-1}$", as shown in Lines 134, 246, 249, 300, 337 and 423 in the revised manuscript and in the Supporting Information.

Line 167: The unit of BET surface area of $SiO_2$ should be $m^2$ $g^{-1}$.

**Re:** Thank you. This has been revised in Line 172 in the revised manuscript.

The sentences in Lines 171-174 can be simplified into "The changes in the chemical compositions on mineral oxides in the $SO_2$ uptake process were investigated by the Fourier transform infrared (FTIR) spectrometer (Thermo Nicolet iS50) equipped with an *in situ* diffuse reflectance accessory and a mercury cadmium telluride (MCT) detector".

**Re:** Thank you. The original sentence has been simplified, as shown in Lines 176-179 in the revised manuscript.

Figure 1A: The physical adsorption of $SO_2$ on $SiO_2$ can be quantified according to the integral at the end of the reaction (t = 80-100 min).

**Re:** Thank you. As you suggested, the proportion of the desorbed $SO_2$ during the uptake process can be quantified by dividing the integral of reversible desorption of $SO_2$ ($t = 80–100$ min) into the integral of the $SO_2$ uptake ($t = 20–80$ min), which was calculated to be 95% and 12% in the dark and under irradiation, respectively. This implies that $SO_2$ uptake in the dark was primarily ascribed to the physical adsorption of $SO_2$, while $SO_2$ uptake under irradiation was mainly attributed to chemical processes or irreversible adsorption. This description has been added in Lines 200-205 in the revised manuscript.

Lines 236-237: The bands at 1074 and 1038 $cm^{-1}$ were both ascribed to sulfite. Why did only 1038 $cm^{-1}$ band appear in the dark condition (Figure 2A) and only 1074 $cm^{-1}$ band appeared in the light condition (Figure 2B)?

**Re:** Thank you. The peaks at 1074 and 1038 $cm^{-1}$ have been reassigned. The bidentate sulfate and bisulfate contributed to the bands at 1260 and 1229/1074 $cm^{-1}$, respectively. The bands at 1038 $cm^{-1}$ may be related to the monodentate sulfite. The sulfate bands (1260, 1229 and 1074 $cm^{-1}$) only appeared under irradiation, while the sulfites (1038 $cm^{-1}$) were only detected in the dark. This suggests that light changed the $SO_2$ conversion pathways on $SiO_2$. These descriptions have been modified in Lines 255-257 and 260-262 in the revised manuscript.

Figure 2: In the light condition, the band at 1038 $cm^{-1}$ decreased, while this band increased in the dark condition. Please explain this phenomenon.

**Re:** Thank you. The bands at 1157/1055 $cm^{-1}$ were assigned to the asymmetric stretching of Si–O (Figure S7). Sulfate generated on the surface may interact with $SiO_2$, leading to a decrease in the intensity of peaks (1157/1055 $cm^{-1}$). The bands (1038 $cm^{-1}$) increased under dark condition, suggesting the formation of sulfites. The related description has been added in Lines 262-265 in the revised manuscript.

Figure S7 and S9: I didn't observe any new peaks at 974 $cm^{-1}$.

**Re:** Thank you. The peak (974 cm$^{-1}$) has been deleted in Figure S7 and S9.

Lines 290-291: The band at 1074 cm$^{-1}$ should be marked in Figure S9.

**Re:** Thank you. The band (1074 cm$^{-1}$) has been marked in Figure S9.

Figure 5: The assignment of 1300 cm$^{-1}$ should be given.

**Re:** Thank you. The band at 1300 cm$^{-1}$ should be ascribed to the sulfate. This description has been added in Line 338 in the revised manuscript.

Figure 5: Less sulfites were formed on kaolinite and $Al_2O_3$, while abundant sulfites were observed on MgO and CaO. Please explain this phenomenon.

**Re:** Thank you. The solubility and effective Henry's law constant of $SO_2$ were positively dependent on pH. Thus, $SO_2$ was more liable to be dissolved to form $HSO_3^-/SO_3^{2-}$ on more alkaline surface, leading to a strong $SO_2$ uptake in the dark (Figure 4A and 4B), and abundant sulfites on surfaces (Figure 5). Nevertheless, gaseous $SO_2$ tends to be adsorbed on kaolinite and $Al_2O_3$ due to less solubility of $SO_2$ on these surfaces, and then converted to sulfate under irradiation (Figure 6). Accordingly, a strong promotion effect of light on $SO_2$ uptake was observed on neutral and weakly alkaline surfaces (Figure 4B). This discussion has been added in Lines 405-411 in the revised manuscript.

**Reviewer 2:**

Unexpectedly photoenhanced sulfates formation by the heterogeneous uptake of $SO_2$ on non-photoactive mineral dust by Han et al.

Summary

This manuscript investigated heterogeneous photochemical reactions of $SO_2$ on five mineral oxides (mainly $SiO_2$, but also kaolinite, $Al_2O_3$, MgO, and CaO) that have no photocatalytic activity, and they found light can significantly enhanced the uptake of $SO_2$ which converts to sulfate. Light intensity, RH, $O_2$ contents and basicity of mineral oxides play key roles in this interfacial chemistry, especially regulates $SO_2$ uptake coefficient. The experiments were

performed under various conditions, i.e. using flow tube reactor to obtain the $SO_2$ uptake kinetics, and DRIFTS measurements to confirm sulfate formation.

The technical part seems sound, and I enjoyed reading the manuscript as it is quite easy to follow. Overall, the whole paper is displayed in good quality, with clear writing and nice figures. I would recommend this manuscript to be published in ACP after considering the following major concerns if these comments are helpful for improving the manuscript.

**Re:** Thank you for your comments.

**Major concerns**

(1) Can any possible contamination or impurity in these non-photoactive mineral dust samples be ruled out in this study? The absorption spectra in Fig. S1 seem clean, but tiny amounts of photoactive components if existed as impurity could make it very different. This is a main worry from my side, in case other people cannot repeat the results.

**Re:** Thank you. The purities of different mineral substances are 95%–98%. If photoactive impurities mainly contributed to the $SO_2$ uptake in the experiment, the $SO_2$ uptake coefficient on impurities should be 20–50 times higher than the current $SO_2$ uptake coefficient and range from $10^{-5}$ to $10^{-3}$. The $SO_2$ uptake coefficient on photoactive substances was reported to be $10^{-7}$–$10^{-6}$ in previous studies (Ma et al., *J. Phys. Chem. A.*, 2019, 123, 1311-1318; Park et al., *Environ. Sci. Technol.*, 2017, 51, 9605-9613). Thus, the impurities in minerals were less likely responsible for the $SO_2$ uptake. This discussion has been added in Lines 219-225 in the revised manuscript.

(2) I am not sure whether this study is the first time to look at $SO_2$ photochemical uptake on non-photoactive mineral surface. Is there any $SO_2$ photochemistry reported on non-photoactive mineral oxides in previous literatures? Does it really unexpected as it seems occur on any surfaces that concluded by the authors, even onto flow tube wall in the blank experiment as shown in Fig. S4-S5?

**Re:** Thank you. The photoenhanced $SO_2$ uptake on non-photoactive surfaces is unexpected, since the previous work mainly focused on photoactive surfaces. Few studies observed the

photochemical uptake of $SO_2$ on $SiO_2$. For example, the photochemical uptake of $SO_2$ (60 ppb) on $SiO_2$ was performed in the blank experiment, accompanied by a certain amount of $SO_4^{2-}$ formation (Zhang et al., *Nature Commun.*, 2022, 13, 5364).

(3) Line 52: Some recent findings on multiphase $SO_2$ oxidation leading to sulfate formation should also be mentioned here. Please read: Liu T, Chan A W H, Abbatt J P D. Multiphase oxidation of sulfur dioxide in aerosol particles: implications for sulfate formation in polluted environments[J]. Environmental Science & Technology, 2021, 55(8): 4227-4242.

**Re:** Thank you. According to your suggestion, some recent literatures have been cited in Lines 53-54 in the revised manuscript.

(4) Line 60-62: "Thus, investigating the heterogeneous oxidation of $SO_2$ on mineral dust is of fundamental importance to reveal large missing sources of atmospheric sulfates in the haze periods." I feel this is likely overstated which may not objectively reflect current understanding.

**Re:** Thank you. According to your suggestion, the sentence has been revised into "Thus, investigating the heterogeneous oxidation of $SO_2$ on mineral dust can provide basic data for the model calculation to evaluate atmospheric sulfates." in Lines 60-62 in the revised manuscript.

(5) Line 97-99: This UV-Vis measurement (300-800 nm) does not match the results shown in Fig. S1.

**Re:** Thank you. Figure S1 has been corrected in the revised Supporting Information.

(6) Fig. S4-S5 shows a blank example for $SO_2$ loss onto the flowtube wall at a specific condition with irradiation. The $SO_2$ uptake coefficient is actually measured following a blank subtraction. Does this blank change at different conditions, i.e. different $O_2$, RH, light intensity?

**Re:** Thank you. Different $O_2$ content, relative humidity and light intensity can change the $SO_2$ uptake in an empty flow tube. Thus, the loss of $SO_2$ on the internal wall of the reactor in blank experiments was carried out under various conditions (Figure S5 as an example), and it has

been deducted for the $\gamma$ calculation. This description has been modified in Lines 143-145 in the revised manuscript.

(7) Line 193-194: Here it says the measurements on non-photoactive mineral oxides are comparable ($10^{-7}-10^{-6}$) with those previously reported in literatures, especially dust containing photocatalytic components. Does this mean both photoactive and non-photoactive mineral oxides showing equal/comparable ability of $SO_2$ photochemical uptake, and those photocatalytic components (such as $TiO_2$, GDD, ATD) do not actually play much role?

**Re:** Thank you. It should be pointed out that the similar uptake coefficient did not mean the comparable ability of photoactive and non-photoactive mineral oxides to $SO_2$ uptake, since the uptake coefficient was highly dependent on environmental conditions ($SO_2$ concentration, relative humidity, mineral oxides mass, light source and pressure) and reactor type (chamber and flow tube reactor), and the uptake coefficients mentioned here were not obtained under the exact same reaction conditions used in our study. This discussion has been added in Lines 214-219 in the revised manuscript.

(8) Line 295-302: The dependence of $\gamma$ on five different minerals is very interesting, and explained by their pH differences. Did the authors check such pH-dependence for the same type of mineral oxides (i.e. $SiO_2$) to really prove this pH effect, i.e. via experimentally adjusting the pH such as adding NaOH?

**Re:** Thank you. The experiments for the pH dependence on $SiO_2$ have been also performed (Figure S11). The pH of $SiO_2$ suspension was adjusted to pH = 9, and $\gamma_{s,\ BET}$ and $\gamma_{s,\ Light}/\gamma_{s,\ Dark}$ were determined to be $(8.79 \pm 0.85) \times 10^{-6}$ and 1.31, respectively. These results suggest that light can generally enhance the $SO_2$ uptake on minerals at a wide pH range. This description has been added in Lines 326-330 in the revised manuscript.

(9) Fig. 4 and Fig. S10: The photo-enhanced $SO_2$ uptake is not very significant for other three minerals, especially CaO. This suggests that the enhanced $SO_2$ photochemical uptake at higher pH (more basic mineral oxides) is actually attributed to $SO_2$ dark uptake, which is a bit

contradict with the pH explanation. Why $SO_2$ dark uptake is so strong under these basic mineral surfaces? Fig. 5 also shows a lot of interesting results but not yet discussed in details. I would suggest the authors to stress the $SO_2$ dark uptake on some basic minerals as an important process, with more detailed discussion here.

**Re:** Thank you. The solubility and effective Henry's law constant of $SO_2$ were positively dependent on pH. Thus, $SO_2$ was more liable to be dissolved to form $HSO_3^-/SO_3^{2-}$ on more alkaline surface, leading to a strong $SO_2$ uptake in the dark (Figure 4A and 4B), and abundant sulfites on surfaces (Figure 5). Nevertheless, gaseous $SO_2$ tends to be adsorbed on kaolinite and $Al_2O_3$ due to less solubility of $SO_2$ on these surfaces, and then converted to sulfate under irradiation (Figure 6). Accordingly, a strong promotion effect of light on $SO_2$ uptake was observed on neutral and weakly alkaline surfaces (Figure 4B). This discussion has been added in Lines 405-411 in the revised manuscript.

(10) Line 347-358: I like these DRIFTS experiments designed by adding $Ru(bpy)_3(Cl)_2$) or $NaHCO_3$. How are these added? Are these $^3SO_2$ or OH scavengers also performed in the flowtube reactor to check the $SO_2$ photochemical uptake, which should be unchanged in the presence of these scavengers?

**Re:** Thank you. To verify the role of intermediate, $Ru(bpy)_3(Cl)_2$) and $NaHCO_3$, acting as the $^3SO_2$ and •OH scavengers, respectively, were mixed with $SiO_2$ powder in an agate mortar, and the mixture was put in the reaction cell of DRIFTS. This description has been added in Lines 184-187 in the revised manuscript. In the flow tube experiments, $SO_2$ uptake would occur on $Ru(bpy)_3(Cl)_2$) and $NaHCO_3$ to form the adsorbed $SO_2$ or sulfite, which could change the value of $SO_2$ uptake coefficient. Thus, the reaction of $SO_2$ with $SiO_2$ in the presence of scavengers was not performed in the flow tube reactor.

(11) Line 368-369: Did you test $SO_2$ uptake coefficient under visible light i.e adding an optical filter at 400 nm? Does visible light (>400 nm) also contribute this photoenhanced $SO_2$ uptake?

**Re:** Thank you. The $SO_2$ uptake experiment in the dark and the visible light (>420 nm) was carried out (Figure S13). An ignorable difference was observed for the $SO_2$ concentration with

or without visible light, suggesting that visible light had a minor contribution to the photoenhanced $SO_2$ uptake. This description has been added in Lines 397-400 in the revised manuscript.

(12) Line 370-372 "It means that any surfaces, providing absorptive sites for $SO_2$, can significantly enhance the photooxidation of $SO_2$ to sulfates." This could be true, but I am afraid it is not very strong yet, especially the current experiments on some basic minerals indicate $SO_2$ dark uptake is more important under these conditions.

**Re:** Thank you. According to the experimental results, some surfaces, providing absorptive sites for $SO_2$, can enhance the photooxidation of $SO_2$ to sulfates. However, the promotion effect would vary with different substances. For example, the current experiments on some basic minerals indicate that light plays a minor enhancement role in the $SO_2$ uptake (Figure 4), but it could still enhance the sulfate formation (Figure 5). This description has been modified in Lines 401-405 in the revised manuscript.

(13) Line 386: The lifetime of $SO_2$ photochemical loss on minerals was calculated and compared with those from literatures. Are these conditions comparable? Otherwise should be very careful.

**Re:** Thank you. The reaction conditions in this study and those in the literatures are different in some respects, and the previously reported $SO_2$ uptake coefficient had a lower value ($10^{-7}$–$10^{-6}$). The huge difference in the $\tau$ of $SO_2$ was also ascribed to the variation in the surface area density. The content of $TiO_2$ in mineral dust was only about 1%, and thus the surface area density of $TiO_2$ was about $10^{-7}$ $cm^2$ $cm^{-3}$, leading to a longer $\tau$ (54 years) for $SO_2$ on $TiO_2$. This discussion has been added in Lines 427-432 in the revised manuscript.

(14) Line 393-416 & Table S1: I have greatest concerns about the last section on atmospheric implications. The importance of this $SO_2$ photochemical chemistry on sulfate budget is not yet strictly evaluated, which needs to be done under a uniform model framework. The current calculation on sulfate production rates and comparison among these mechanisms are still very

speculative, based on my opinion. Thus, it should be not extrapolated too much. I would suggest to minimize these text and reservedly conclude that this is a newly identified sulfate formation pathway that might occur in some dust-rich conditions.

Re: Thank you. According to your suggestion, the comparison of the sulfate budget between the results in previous work and that in our study (Table S1 and relevant description in the manuscript) has been deleted. A moderate conclusion of this study has been given in Lines 450-451 in the revised manuscript.

**Minor comments**

Line 39: change to "… with the mass fraction of sulfates … "

Re: Thank you. This has been modified in Lines 38-39 in the revised manuscript.

Line 114: How did the Reynolds number ($Re$) being calculated?

Re: Thank you. The calculation of Reynolds number ($Re$) has been added in the revised Supporting Information.

Line 142: Can you provide the detailed numbers (i.e., $V$, $S$, $w$, $D$, $N_u$, etc) you used for equation (2), (3) and (4) calculations?

Re: Thank you. The detailed numbers in equations (2), (3) and (4) have been added in Lines 149, 157 and 171-173 in the revised manuscript.

Line 151: "The corrected $\gamma$ can be calculated by inserting the equation 3 into the equation 2". I am a bit confused here. My understanding is that equation (3) is to give a corrected $k$, then it needs a separate equation to calculate corrected $\gamma$.

Re: Thank you. The description for the calculation of the corrected $\gamma$ has been modified. The corrected $\gamma$ can be calculated by the equation 2 where $k$ was replaced by $k'_{r, SiO_2}$. The details have been modified in 148-160 in the revised manuscript.

Line 186: didn't = did not

**Re:** Thank you. This has been modified in Line 197 in the revised manuscript.

The light intensity in many places are presented i.e. 250W/m$^2$ or xxx photons cm$^{-2}$ s$^{-1}$. I am not sure they are the same

**Re:** Thank you. The unit (W/m$^2$ and photons cm$^{-2}$ s$^{-1}$) represents the light intensity, which has been unified into photons cm$^{-2}$ s$^{-1}$ in Lines 134, 246, 249, 300, 337 and 423 in the revised manuscript and Supporting Information.

Thank you very much for your help.

Sincerely yours,

Chong Han

\*\*\*\*\*\*\*\*\*\*\*\*\*\*\*\*\*\*\*\*\*\*\*\*\*\*\*\*\*\*\*\*\*\*\*\*\*\*\*\*\*\*\*\*\*\*\*\*\*\*\*\*\*\*\*\*\*\*\*\*\*\*\*\*\*\*\*\*\*\*\*\*\*\*\*\*\*\*\*

Professor Chong Han

School of Metallurgy

Northeastern University

Shenyang 110819, China

E-mail: hanch@smm.neu.edu.cn

---

## Author Response (AR2)

Dear professor Thorsten Bartels-Rausch,

Thank you for your decision to allow us to revise our manuscript for publication in *Atmospheric Chemistry and Physics*. Below is a point by point response to the comments.

**Referee #1:**

Thanks for the detailed response from the authors. The authors have clarified most of my comments and made significant improvement in this revision. There is still one concern related to the possible contamination or impurity of trace photocatalytic components in these non-photoactive mineral dust samples, which is NOT yet ruled out but is indeed my main worry in my comment (1).

The authors reported the purities of different mineral substances to be 95%–98%. They argued that if their observation is driven by photoactive impurity, then $SO_2$ uptake on photoactive substances would be a factor of 20-50 higher than 1E-7 to 1E-6, while $SO_2$ uptake coefficient on photoactive substances was normally reported to be 1E-7 to 1E-6 in previous literatures.

This is clearly not a good argument because some studies did not use pure photoactive substances, e.g. Arizona Test Dust (ATD) as cited in Park et al., 2017, ES&T. I found it is also a bit contradictory in their subsequent reply to my comment (7): "Does this mean both photoactive and non-photoactive mineral oxides showing equal/comparable ability of $SO_2$ photochemical uptake, and those photocatalytic components (such as $TiO_2$, GDD, ATD) do not actually play much role", where they explained due to different specific experiment conditions. Such explanation is always true, but again, it is still necessary to rule out the possible photoactive impurity used in their samples through detailed chemical characterization.

To be rigorous and cautious, I would encourage the authors to provide a table containing chemical analysis and elemental distribution of their samples, if applicable prior published in ACP. A good example is shown in the SI Table S1 in Dupart et al., 2012, PNAS. Another paper cited by the authors, e.g., Park et al., 2017, ES&T, also provides mineral dust chemical characterization in SI Section S1. In addition, the newly added text in both comments (1) and (7) should be rephrased to minimize such logical contradicts. The new data ($SO_2$ uptake on $SiO_2$ at pH=9) in Figure S11 is good, which can be implemented in the main text Figure 4.

Refs

Dupart Y, King S M, Nekat B, et al. Mineral dust photochemistry induces nucleation events in the presence of $SO_2$[J]. Proceedings of the National Academy of Sciences, 2012, 109(51): 20842-20847.

Park J, Jang M, Yu Z. Heterogeneous photo-oxidation of $SO_2$ in the presence of two different mineral dust particles: Gobi and Arizona dust[J]. Environmental science & technology, 2017, 51(17): 9605-9613.

**Re:** Thank you for your comments. According to your suggestion, the chemical composition of $SiO_2$ has been analyzed, as shown in Table S1 of the revised Supporting Information. Table S1 shows that the fraction of $SiO_2$ in the sample was 99.02%, accompanied by a small amount of $Al_2O_3$, $K_2O$, $Fe_2O_3$ and $CaO$. Photoactive substances ($Fe_2O_3$) was very few in the sample, and they should not be the main contributor to the photochemical uptake of $SO_2$. This description has been added in Lines 218-221 in the revised manuscript, which avoided a bit contradictory in original comment (1) and (7).

The new data ($SO_2$ uptake on $SiO_2$ at pH=9) in Figure S11 has been implemented in Figure 4 in the revised manuscript.

**Referee #3:**

Despite decades of research on the oxidation of $SO_2$ to sulfate in the atmosphere, it is surprising that some potential pathways have still not been studied in sufficient detail. Yang et al. provide such a study with detailed experiments shedding light on the heterogeneous oxidation of $SO_2$ on non-photoactive dust. They focus on $SiO_2$, but also present results for other dust types, which together actually constitute by far the largest fraction of mineral dust in the atmosphere. This topic is, thus, of interest for the readers of ACP. As acknowledged by the other reviewers the study is sound and clearly structured. It now also contains all relevant technical information and I recommend publishing it. Nevertheless, I add below some comments, which only concern editorial or technical corrections or improvements, which should be considered before publication.

**Re:** Thank you for your comments.

**Major comment**

I would like to come back to one of the comments by reviewer 2, who wondered if other studies on $SO_2$ oxidation on non-photoactive minerals exists. In their response the authors admit that such studies can be found mentioning the study by Zhang et al.. Other studies can also be cited like Xu et al. (Sulfur isotope composition during heterogeneous oxidation of $SO_2$ on mineral dust: The effect of temperature, relative humidity, and light intensity, Atmospheric Research, 254, 2021, 105513, https://doi.org/10.1016/j.atmosres.2021.105513) and should be mentioned in the manuscript. Taking this into account, I still think that statements like (L. 347) "Nevertheless, it was very surprised that the light can greatly promote the formation of sulfates via the $SO_2$ uptake process on mineral oxides without photocatalytic activity, which was strongly suggested to be a new and important finding for atmospheric sulfate sources." or (L. 449) "This suggests that the $SO_2$ on non-photoactive surfaces is a newly identified sulfate formation pathway in some dust-rich conditions." should still be toned down.

**Re:** Thank you. Few studies observed the photochemical uptake of $SO_2$ on non-photoactive minerals (Xu et al., 2021; Zhang et al., 2022). This description has been added and the related studies were cited in Lines 196-197 in the revised manuscript. The original sentences in Lines 347-350 and 449 have been toned down. Sentences have been revised into "Nevertheless, the light can greatly promote the formation of sulfates via the $SO_2$ uptake process on mineral oxides without photocatalytic activity", and "This suggests that the $SO_2$ uptake on non-photoactive surfaces may be an important sulfate formation pathway under irradiation in some dust-rich conditions." in Lines 342-344 and 442-443 in the revised manuscript, respectively.

In my opinion the manuscript requires some serious proofreading and editing since it still contains a high number of language errors or confusing or incomplete sentences. Some examples from the introduction are:

L. 35: "Typical mixing ratios of $SO_2$…"

L. 36: "…for a clean weather in remote areas,…"

L. 38: "…which is one of the most significant compositions in fine particles.

L. 38: "The mass fraction of sulfates in $PM_{2.5}$ is high up to 30% …"

L. 43: "…causing respiratory illness and cardiovascular (…"

I realized that one of the sentences was recommended by one of the reviewers (L. 38), but still

it should be re-written.

**Re:** Thank you. These original sentences have been modified, as shown in Lines 35, 36, 38-39 and 42-43 in the revised manuscript. We also carefully revised the full text to avoid other mistakes.

L. 327ff: The uptake coefficients on $SiO_2$ with a pH adjusted to 9 are mentioned. It would be good to include these numbers as further bars in the Figures 4A and 4B.

**Re:** Thank you. According to your suggestion, the data ($SO_2$ uptake on $SiO_2$ at pH=9) in Figure S11 has been implemented in Figure 4A and 4B in the revised manuscript.

Fig. 5: Wouldn't it be better to arrange all experiments in the dark in the top row and all experiments with irradiation in the bottom row?

**Re:** Thank you. According to your suggestion, the experimental results (Figure 5) in the dark and with irradiation have been arranged in the top and bottom rows, respectively.

Thank you very much for your help.

Sincerely yours,

Chong Han

\*\*\*\*\*\*\*\*\*\*\*\*\*\*\*\*\*\*\*\*\*\*\*\*\*\*\*\*\*\*\*\*\*\*\*\*\*\*\*\*\*\*\*\*\*\*\*\*\*\*\*\*\*\*\*\*\*\*\*\*\*\*\*\*\*\*\*\*\*\*\*\*\*\*\*\*\*\*\*\*\*\*

Professor Chong Han

School of Metallurgy

Northeastern University

Shenyang 110819, China

E-mail: hanch@smm.neu.edu.cn

---

## Author Response (AR3)

Dear professor Thorsten Bartels-Rausch,

Thank you for your decision to allow us to revise our manuscript for publication in *Atmospheric Chemistry and Physics*. Below is a point by point response to the comments.

Dear Chong Han

Thank you for your patience, understanding, and response to the referee's comments. I would like to come back to the "tone down." Thank you for doing the in the sentences mentioned by the referee. I understood and supported the referee in that those were just examples. In the current version, for example, line 7 still states "prominently": "We provide evidence that light prominently enhances the conversion of $SO_2$". Could you revise the manuscript accordingly throughout? What is the meaning of prominently that you refer to? Can you give statistical measures to support this statement? I failed to find a discussion in the manuscript that supports the "prominent" chemistry. Do you refer to the atmospheric implications or the signal difference in Fig 1A?

**Re:** Thank you for your comments. "prominently" in the manuscript refers to the signal difference in Fig 1A, and we statistically described the $SO_2$ uptake in the dark and under irradiation, as shown by the sentences "The $\gamma_{0,\,BET}$ and $\gamma_{s,\,BET}$ under the irradiation of $7.93 \times 10^{16}$ photons $cm^{-2}$ $s^{-1}$ were 1.75 and 2.25 times of those in the dark, respectively." in Lines 208–210 in the revised manuscript. According to your suggestion, the descriptions of the results and the significance of the work have been toned down throughout the manuscript. For example, the words "prominently", "significantly", "firstly", "greatly", "strong", "obviously" and "markedly" have been deleted in the revised manuscript.

The language aspects that the referee referred to can be dealt with during copy editing (https://publications.copernicus.org/services/copy_editing_for_english.html)

I hope this helps and looking forward to your revised version.

best, Thorsten Bartels-Rausch

**Re:** Thank you. We will carefully revise the language with the help of the copy-editing team in the proofreading stage.

Thank you very much for your help.

Sincerely yours,

Chong Han
* * *
Professor Chong Han

School of Metallurgy

Northeastern University

Shenyang 110819, China

E-mail: hanch@smm.neu.edu.cn